# Parasitic Infections in Stranded Whales and Dolphins in Canary Islands (2018–2022): An Update

**DOI:** 10.3390/ani14233377

**Published:** 2024-11-23

**Authors:** Zuleima Suárez-González, Jorge F. González, Manuel Arbelo, Eva Sierra, Ayoze Castro-Alonso, Julia N. Hernández, Vidal Martín, Natalia Fraija-Fernández, Antonio Fernández

**Affiliations:** 1Division of Histology and Veterinary Pathology, Atlantic Center for Cetacean Research, University Institute for Animal Health and Food Safety (IUSA), Veterinary School, University of Las Palmas de Gran Canaria (ULPGC), Transmontaña, Arucas, 35413 Canary Island, Spain; zuleima.suarez@ulpgc.es (Z.S.-G.); manuel.arbelo@ulpgc.es (M.A.); eva.sierra@ulpgc.es (E.S.); ayoze.castro@ulpgc.es (A.C.-A.); antonio.fernandez@ulpgc.es (A.F.); 2University Institute for Animal Health and Food Safety (IUSA), Veterinary School, University of Las Palmas de Gran Canaria (ULPGC), Transmontaña, Arucas, 35413 Canary Island, Spain; julia.hernandez@ulpgc.es; 3Society for the Study of Cetacean in the Canary Archipelago (SECAC), Arrecife, Lanzarote, 35500 Canary Islands, Spain; vidal@cetaceos.org; 4Marine Zoology Unit, Cavanilles Institute of Biodiversity and Evolutionary Biology, University of Valencia, 46980 Valencia, Spain; natalia.fraija@uv.es

**Keywords:** cetaceans, conservation, parasites, identification, health

## Abstract

Cetaceans are considered indicator species for the health of the oceans. These wildlife animals are often parasitized by a wide range of species, including external and internal parasites. In turn, the parasites may provide valuable information in biological and ecological terms, where parasitic identification is a fundamental issue for etiological diagnosis, disease epidemiology and health assessment. Here, we update the record of parasites, by morphological identification, that are present in cetaceans stranded in the Canary Islands. Up to eighteen species and nine genera were identified in a wide range of hosts, both dolphins and whales, with different prevalence in each case. These data contribute and improve the general knowledge of the parasitic fauna in Canarian cetaceans, which requires special attention as an important source of information on host ecology and biology, as well as on the health status and imbalances that occur in the parasite–host association.

## 1. Introduction

Marine mammals in the Canary Islands, particularly cetaceans, serve as invaluable bioindicators of the marine ecosystem’s environmental status, providing insights into pollution levels and ecological shifts [1,2,3]. Their health is impacted by various stressors, including parasitic infections [4,5,6,7], which may involve internal parasites (such as protists, nematodes, cestodes, trematodes, and acanthocephalans) and external parasites like crustaceans [8,9,10].

Parasites exert a substantial influence on the health of marine mammals and provide critical insights from both biological and ecological perspectives. They are considered indicators of health at the individual, population, and environmental levels [11,12,13]. Although parasitism **is** common in wildlife, host–parasite interactions can become unbalanced, leading to negative effects that range from debilitation to mortality and, in some instances, contribute to stranding events [14,15,16,17]. The severity of these effects depends on several factors, such as the host’s immune status, parasite species and burden and the presence of additional pathogens [13,18,19].

Although few marine mammal parasites exhibit high pathogenicity, those with significant prevalence have been implicated in strandings and mortality [8,14,17]. For instance, worms such as the genus *Crassicauda* Leiper and Atkinson, 1914 [14,20,21,22] and various Pseudaliids Railliet & Henry, 1909 were found in the lungs and the pterygoid sacs [15,23]. Moreover, trematodes of the genus N*asitrema* Ozaki, 1935, have been observed in these sacs, which may be implicated in severe central nervous system lesions, directly leading to stranding and death [16,24,25,26,27,28]. In addition, other hepatic and pancreatic trematodes have been identified in a wide range of host species [29,30,31,32,33]. Some of these parasites in cetaceans have an important zoonotic relevance, posing risks to human health and the economy [3,12]. Notable examples include the nematode genus *Anisakis* Dujardin, 1845 and the zoonotic protist *Toxoplasma gondii* Nicolle & Manceaux, 1908 [34,35,36,37]. While cetaceans are not typically consumed directly by humans in most regions, infections can be associated with the ingestion of larvae through the consumption of raw or undercooked fish [38]. Additionally, cetaceans play an important role as potential reservoirs and sentinels for zoonotic parasitic agents, contributing to their maintenance and distribution within aquatic environments and participating in the natural transmission cycles of parasites and other infectious agents transmissible to humans [39,40].

Historically, research on the parasitic fauna in marine mammals has been limited, with most studies focusing on parasites found in stranded carcasses, often in advanced stages of decomposition, which complicates analysis [8,18,19,41,42]. Despite these challenges, this manuscript provides new records of parasite species and host interactions in cetaceans from the Canary Islands, contributing novel insights into previously undescribed associations in the region. In total, 31 different species of cetaceans, including 24 odontocetes and 7 mysticetes, have been identified in the Canary Islands by 2022 [43].

The accurate morphological identification of parasites infecting these marine mammals is essential for determining the etiology of parasitic diseases, conducting epidemiological investigations and assessing the health status of both individual animals and their populations [5]. In this study, we updated the genera and parasitic species that affect cetaceans in the Canary Islands, including odontocetes and mysticetes stranded between 2018 and 2022.

## 2. Materials and Methods

### 2.1. Necropsied Animals and Biological Data

Necropsies were performed on 233 stranded animals in the Canary Islands during the period 2018–2022. The necropsy protocol was carried out according to standard and innovative technological procedures [5,6]. Biological data, including species identification, biometrics, age category, sex and body condition, were recorded (Table 1). Age ranges were based on total body length and gonadal development, with individuals categorized as fetus/neonate, calf, juvenile, subadult or adult [5,6,44]. Body condition was assessed according to anatomical parameters and classified into four categories: good, moderate, poor and very poor [45].

### 2.2. Parasite Analysis

Data on the parasite occurrence were obtained from the necropsy reports for all 233 animals examined. Each case included in this study was diagnosed through routine pathological and cause-of-death analyses of stranded cetaceans, performed at the Division of Histology and Animal Pathology, Institute for Animal Health (IUSA), within the Veterinary School at the Universidad de Las Palmas de Gran Canaria. Tissue samples were systematically collected during necropsy, fixed in 10% neutral buffered formalin, carved, routinely processed, embedded in paraffin, sectioned at 5 μm and stained with hematoxylin and eosin (HE) for light microscopic examination (Olympus BX51, Tokyo, Japan). Imaging was conducted using Camera software for DP21 (Ver.02.01.01.93) for pictures or figures (Olympus DP21, Tokyo, Japan). Direct parasite collection, however, was only possible for 57 animals stranded between 2020 and 2022. Complete individual parasites from these cases were preserved in 70% ethanol for subsequent laboratory morphological identification. Nematodes and acanthocephalans were cleared with a lactophenol solution to facilitate better observation of internal structures. Cestodes and trematodes were initially stained with acetocarmine, then dehydrated through a graded ethanol series, cleared with dimethyl phthalate and mounted in a permanent preparation using Canada balsam, following the methods previously used [46]. For detailed observation, a magnifying glass and, primarily, an optical microscope equipped with a monitor were employed. Motic^®^ Images Plus 2.0 ML software and a Moticam digital camera were used to measure parasite length in μm. Parasites were morphologically identified using specific taxonomic guides and keys [25,47,48,49,50,51,52,53,54,55,56,57,58].

Microscopic protists were identified in histopathological tissue sections, with particular focus on tissues and organs known to harbor *Sarcocystis* (Lankester, 1882) and *Toxoplasma gondii* cysts [26,59,60]. *Toxoplasma gondii* was identified molecularly using a recently developed real-time PCR method described by Sierra et al. [26]. The PCR products obtained from positive cases were processed and sequenced according to Sierra et al. [26], and a BLAST search was performed to verify the identity of the PCR amplicons.

## 3. Results

### 3.1. Necropsied Animals and Biological Data

In the period 2018–2022, 233 animals were necropsied in the Canary Islands belonging to 18 different cetacean species. Animals in a very advanced state of decomposition were not included in this study. In total, 61 striped dolphins (*Stenella coeruleoalba)*, 51 Atlantic spotted dolphins (*Stenella frontalis*), 29 common dolphins (*Delphinus delphis*), 17 bottlenose dolphin*s* (*Tursiops truncatus*), 15 short-finned pilot whales (*Globicephala macrorhynchus*), 11 Risso’s dolphins (*Grampus griseus*), 11 pygmy sperm whale*s* (*Kogia breviceps*), 8 sperm whales (*Physeter macrocephalus*), 8 Cuvier’s beaked whales (*Ziphius cavirostris*), 6 rough-toothed dolphins (*Steno bredanensis*), 3 fin whales (*Balaenoptera* physalus), 3 Blainville’s beaked whale*s* (*Mesoplodon densirostris*), 3 Gervais’s beaked whales (*Mesoplodon europaeus*), 2 common minke whales (*Balaenoptera acutorostrata*), 2 Fraser’s dolphins (*Lagenodelphis hosei*), 1 Bryde’s whale (*Balaenoptera edeni*), 1 dwarf sperm whale (*Kogia sima*) and 1 melon-headed whale *(Peponocephala electra*) were analyzed. Stranded necropsied cetaceans and their respective biological data and stranding location are presented in Appendix A.

### 3.2. Parasite Analyses

Among the 233 animals necropsied, 192 (82%) were found to be parasitized by one or more species, including nematodes, trematodes, cestodes, acanthocephalans, crustaceans and protists. From the total parasitized animals, parasites were only obtained from 30% of the animals, and 18 parasitic species were identified. Many parasites could not be identified at the species level due to the poor condition of the preservation of the specimens. Table 1 shows the prevalence of parasitism among animals, categorized by sex, age and body condition. Generally, similar prevalence was observed between female and male animals, with the exception of sperm whales, likely due to the higher number of female than male strandings. In addition, adult and juvenile/sub-adult animals showed a higher parasitism rate among species with the most frequent strandings, such as the *D. delphis*, *S. frontalis*, *S. coeruleoalba* and *T. truncatus*. Furthermore, variable differences were observed in the body condition of these animals.

#### 3.2.1. Nematodes

The nematode genus *Crassicauda* had a global prevalence of 43%. The highest prevalence was observed in *Z. cavirostris* (100%) and *S. frontalis* (61%) (Table 2). In *Z. cavirostris*, *Crassicauda anthonyi* Chabaud, 1962 was morphologically identified as responsible for renal parasitosis and vascular damage (Figure 1A), and *Crassicauda grampicola* Johnston & Mawson, 194 was identified parasitizing the pterygoids sacs in two *S. frontalis*, one *T. truncatus* and four *G. griseus* (Figure 1B). *Crassicauda* sp. was also found in the mammary glands, prostate and subcutaneous tissue of several odontocete species and one mysticete (*B. edeni*) (Table 3).

Additionally, the lungs of various stranded odontocetes were frequently parasitized, with the highest prevalence observed in *T. truncatus* (88%) and *D. delphis* (52%) (Table 2). *Halocercus delphini* Baylis & Daubney, 1925 was morphologically identified in two *S. coeruleoalba* (Figure 1C), while *Stenurus ovatus* (von Linstow, 1910) Baylis & Daubney, 1925 was found in four *T. truncatus*, diagnosed by its characteristic cuticle inflammation (Figure 1D). Although not identified at the species level, *Halocercus* sp. and *Stenurus* sp. were also detected in one *S. frontalis* and *S. coeruleoalba*, respectively. Nematodes in the pterygoid sacs were observed in 13% of the animals, with the highest prevalence in *G. macrorhynchus* (67%) (Table 2). *Stenurus globicephalae* Baylis et Daubney, 1925 was the only nematode species identified in the pterygoid sacs of three *G. macrorhynchus* and one *G. griseus* (Image 1e). The zoonotic nematode *Anisakis* sp. had an overall prevalence of 35% across 14 different odontocete species (Table 3), with the highest frequency observed in *K. breviceps* (72%) (Table 2). Both the dwarf sperm whale and melon-headed whale had a 100% prevalence of anisakiasis, although the number of strandings for these species was low (*n* = 1 for each). Adults of *A. simplex* were identified in one *D. delphis* (Figure 1F) and in the *P. electra*, leading to ulcerated areas. In addition, a species likely to be *A.* conf. *physeteris* was found in a *K. breviceps*, but confirmation was not possible due to the decomposition state of the carcass and the parasites.

#### 3.2.2. Trematodes

The prevalence of *Nasitrema* spp. was 23%, predominantly affecting the pterygoid sacs and occasionally the central nervous system (CNS) in odontocetes (Table 3). The highest prevalence was observed in *T. truncatus* (71%) and *S. bredanensis* (50%) (Table 2). *Nasitrema delphini* Neiland, Rice, & Holden, 1970 was morphologically identified in three *T. truncatus*, two *S. coeruleoalba* and one *D. delphis* (Table 2 and Figure 2A).

Hepatic trematodes were found in 23% of the stranded marine mammals, primarily affecting the bile ducts, while pancreatic trematodes were detected in 24% of the animals. Varied prevalence was observed for liver flukes, with *D. delphis* and *G. macrorhynchus* showing higher infection rates in the pancreas. Among the stranded species, *S. coeruleoalba* and *S. frontalis* were the most frequently parasitized by liver and pancreatic trematodes. *Brachycladium atlanticum* (Abril, Balbuena and Raga, 1991) Gibson, 2005 (Figure 2B) was identified in *D. delphis* and three *S. coeruleoalba*, while *Oschmarinella rochebruni* (Poirier, 1886) Gibson & Bray, 1997 was morphologically identified in one *S. bredanensis*, two *S. coeruleoalba*, two *S. frontalis*, one *D. delphis* and one *T. truncatus*, affecting both the bile ducts and pancreas (Figure 2C). Brachycladiid trematodes, with an everted cirrus, compatible with the genus *Oschmarinella* were also collected in the bile ducts of two *Z. cavirostris*; however, their specific identity was not obtained. In addition, cysts of *Pholeter gastrophilus* (Kossack, 1910) Odhner, 1914 were found embedded in the mucosa, mainly in the glandular and pyloric regions of the stomach (Figure 2D), with the highest prevalences observed in *S. coeruleoalba* (59%) and *T. truncatus* (59%) (Table 2).

#### 3.2.3. Cestodes

Cestodes were identified in the intestine of 16% of the stranded cetaceans analyzed in this study. However, only the genus *Diphyllobothrium* Cobbold, 1858 was identified in seven *S. coeruleoalba* (Table 2) due to the advanced decomposition state of these parasites and the absence of scolex and mature proglottids (Figure 3A). In addition, larval forms of the cestodes of *Clistobothrium delphini* (Bosc, 1802) Caira, Jensen, Pickering, Ruhnke & Gallagher, 2020 were detected in 14 different species, mainly in the anogenital region (Figure 3B). *Clistobothrium grimaldii* (Moniez, 1889) Caira, Jensen, Pickering, Ruhnke & Gallagher, 2020 larvae were mainly observed in the abdominal serosa (Figure 3C). Both types of metacestodes were present in mysticetes and odontocetes, with varying prevalence across different host species (Table 2).

#### 3.2.4. Acanthocephalans

Acanthocephalans were detected in only 8% of the stranded animals in seven different odontocete species (Table 3). Specifically, immature specimens of *Bolbosoma vasculosum* (Rudolphi, 1819) Porta, 1908 (Figure 4A) were identified in one *D. delphis* and two *S. coeruleoalba* (Table 2), while *Bolbosoma capitatum* (von Linstow, 1880) Porta, 1908 was identified in four *G. macrorhynchus* and one *P. electra* (Figure 4B). These parasites were embedded by their cephalic end in the gastric and intestinal mucosa.

#### 3.2.5. Crustaceans

Ectoparasitic crustaceans were found in 44% of the stranded cetaceans, including cyamids, barnacles and copepods. Table 3 shows the parasitized host species and their location. Two specimens of the genus *Cyamus* Latreille, 1796 were identified in one *Z. cavirostris* (Figure 5A). Among the barnacles, *Conchoderma auritum* Linnaeus, 1767 was identified on a *Z. cavirostris* (Figure 5B), based on its ear-like extension in the capitulum, and *Conchoderma* sp. in five different species of odontocetes (Table 2). *Xenobalanus globicipitis* Steenstrup, 1852 was identified on eleven *D. delphis*, five *G. macrorhynchus*, four *G. griseus*, one *P. electra*, seven *S. coeruleoalba*, eight *S. frontalis*, one *S. bredanensis* and two *T. truncatus* (Table 2), primarily on the caudal, dorsal and pectoral fins (Figure 5C). At the same time, large-size copepods, *Pennella balaenoptera* Koren & Danielssen, 1877, were seen attached to the skin and embedded on the muscle (Figure 5D) in seven different host species (Table 2 and Table 3).

#### 3.2.6. Protists

Infection with the genus *Sarcocystis* was diagnosed in 8% of the animals based on histological examination, with revealed multifocal cysts within myofibers (Figure 6). No associated inflammatory processes were observed in any of the cases. *Toxoplasma gondii* was detected in two *S. coeruleoalba* during 2018–2022, with a total prevalence of 3% in this odontocete species (Table 2). Systemic toxoplasmosis was associated with an inflammatory reaction, and intracellular protist cysts were observed in the CNS (Figure 6B), as well as in the heart, skeletal muscle, adrenal glands, pancreas, liver, lung, spleen, pituitary, thyroid, genitourinary and digestive tracts through histopathological analysis in both animals. Molecular testing confirmed the presence of *T. gondii* in both animals.

## 4. Discussion

The Canary Islands are recognized for their high natural value, attributed to their oceanographic characteristics, substantial cetacean biodiversity (comprising 31 species) and their strategic location along the migratory routes of numerous seasonal and a few resident cetacean species [43]. Parasitism in wildlife is prevalent, with cetaceans often hosting multiple parasitic species throughout their lives. In addition, these animals may experience infectious processes due to other pathogens, toxic exposure and various natural or anthropogenic factors, which can disrupt the balance in parasite–host interactions [10,18,61]. Anthropogenic activity in the Canary Islands has significantly increased, exposing marine mammals to various threats such as fishing nets, the ingestion of marine litter and vessel collisions [62,63,64]. The interaction between cetaceans and their parasites may have implications for disease transmission to other marine species and humans [65,66]. Therefore, comprehensive research on cetacean-associated parasites is crucial for understanding the ecological dynamics of these populations and for developing effective conservation and management strategies to preserve marine biodiversity in the region while assessing animal health. During the period 2018–2022, 233 cetacean carcasses from eighteen different marine mammal species were necropsied. The stranding of cetaceans in the Canary Islands can result from both natural and anthropogenic causes [5,6,67]. Of the cetaceans examined, 82% were found to be parasitized, with neonates being less frequently parasitized compared to juveniles/sub-adults or adults. This pattern likely reflects the increasing exposure to parasites that wildlife encounters throughout their lives [68]. While parasite prevalence was similar between males and females, in some cases, high parasite burdens were noted, compromising the health of the infected organ and the overall condition of the animal, depending on the parasite involved. However, no clear relationship was observed between parasitism and body condition in this study, which contrasts with findings from other research that have identified such associations [69].

Nematodes identified as *Crassicauda* spp. were associated with higher parasite burdens in this survey and severe lesions on host individuals. Relevant inflammatory responses were observed in the kidneys, muscles, prostate, mammary glands and pterygoid sacs of parasitized cetaceans, consistent with previous studies [20,21,22,70,71,72,73]. Despite the extensive lesions in the kidneys, characterized by a predominant eosinophilic response, parasitism did not alter the body condition. Special attention should be given to the inflammation of the pterygoid sacs, as lesions in this area have been proposed as a potential cause of strandings [74,75]. Nematodes of this genus were highly prevalent in the studied carcasses. Although crassicaudid species have been reported in a wide range of organs and tissues, including the vasculature, urogenital organs, placenta, mammary glands, cranial sinuses, lungs, musculature and subcutaneous fat layer [19], each species appears to exhibit a distinct tissue-specific localization and variability in host specificity. Thus, *Crassicauda anthonyi* appears to be a species-specific parasite of *Z. cavirostris*, primarily affecting the kidneys, as recently reported in Brazil [76], the Mediterranean Sea [77] and Japan [78]. Similarly, *Crassicauda grampicola* is known to have a strong affinity for *G. griseus*, particularly inhabiting the pterygoid sacs, where it can induce local inflammation, potentially contributing to strandings [15,21,79,80]. In our study, *C. grampicola* was identified in the pterygoid sacs of three odontocete species: *G. griseus*, *S. frontalis* and *T. truncatus*. The absence of nematodes in the pterygoid sacs of mysticete cetaceans in our study may be explained by the less frequent examination of their cranial regions, primarily due to logistical constraints. Additionally, this parasite species has been reported in other organs, such as the mammary gland of Atlantic white-sided dolphin (*Lagenorhynchus acutus*) [81] and the prostate gland in four different odontocete species in the Canary Islands [73]. However, in the present study, the identification of *C. grampicola* in reproductive organs was not possible. Finally, *C. boopis* is commonly known to parasitize the urinary tract of *B. physalus* [71,72,77,82], though in this case, the species-level identification of the *Crassicauda* sp. found in the fin whale was not achieved.

Pulmonary nematodes from the family Pseudaliidae are among the most commonly encountered parasites in cetaceans, with species of the genera *Halocercus*, *Stenurus* and *Pseudalius* being frequently reported [83]. In this study, *Halocercus delphini*, *Stenurus globicephala* and *Stenurus ovatus* were identified in four cetacean species. To the authors’ knowledge, this represents the first morphological identification of these species in the Canary Islands. On the northwest coast of Spain, nematodes of the genus *Stenurus* were found to be the most prevalent in the lungs of stranded odontocetes, although they were also observed in the pterygoid sacs [84]. In the present study, *Stenurus globicephalae* was identified in *G. macrorhynchus* and *G. griseus*, while *Stenurus ovatus* and *Halocercus delphini* were exclusively found in *T. truncauts* and *S. coeruleoalba*, respectively. This aligns with the existing literature, which indicates that *Stenurus globicephala* is host-restricted to the clade Globicephalinae, whereas *Stenurus ovatus* and *Halocercus delphini* are associated with the clade Delphininae [84,85]. The findings of this survey support this observation; however, the low prevalence of these nematodes in our study underscores the need for further data to strengthen this hypothesis and provide more comprehensive insights into host–parasite relationships.

*Anisakis* spp. were commonly found in the forestomach of various hosts in this survey, as expected, across fourteen odontocete species. These nematodes were sometimes observed free in the stomach lumen, while in other cases, they were attached to the gastric mucosa. In some instances, inflammatory and ulcerated areas, hemorrhages and necrotic lesions were observed in the stomach, as previously described [36,86,87,88]. Such lesions caused by *Anisakis* spp. have also been documented in the Canary Islands [89]. Unfortunately, as noted in other studies [90], the poor conservation status of many of these worms made species identification difficult. However, *A. simplex* was identified in two animals from our study, a *D. delphis* and a *P. electra*, marking the first recorded instance of adult *A. simplex* in the *P. electra*, to the authors’ knowledge. A clear pattern of host preference has been reported for *Anisakis* simplex, with a predominant infection in delphinids [91]. The presence of this zoonotic nematode in the Canary Islands is of particular importance, as it poses potential risks to the local fishing industry and public health and should be considered in local food safety regulations. Further investigation into the prevalence and impact of *Anisakis* spp. in this region is warranted.

Several trematodes are well-known pathogens in marine mammals, with some species causing significant health impacts. *Nasitrema delphini* has been previously reported, in a stranded *Mesoplodon densirostris* in the Canary Islands, associated with mild to severe sacculitis, neuritis, otitis and meningoencephalitis [27]. In this study, *N. delphini* was also detected in *D. delphis*, *S. coeruleoalba*, *T. truncatus* and, frequently in association with inflammatory lesions. In the liver and pancreas, *Brachycladium atlanticum* was primarily observed, as expected, in *S. coeruleoalba*, in line with previous findings from the Mediterranean Sea [92], and in *D. delphis*. *Oschmarinella rochebruni* was detected in several other members of the Delphinidae family, consistent with reports from other studies worldwide [19,92,93]. Less frequent is the presence of trematodes in the liver of *Z. cavirostris* [48,49,53,54]. Even though the trematode specimens were found in good conditions and some but not all specimens, had an everted cirrus, they were identified within the genus *Oschmarinella*, but specific identification was not possible. A more accurate diagnosis is necessary to confirm the species involve.

*Diphyllobothrium* Cobbold, 1858 is the only zoonotic cestodes known to infect marine mammals [94,95]. Although it has previously been recorded in the Canary Archipelago in *D. delphis*, *S. coeruleoalba* and *S. frontalis* [89]. However, during the five years of this study, it was only identified in *S. coeruleoalba*. Unsurprisingly, cysticercosis caused by *Clistobothrium delphini* and *Clistobothrium grimaldii* was commonly found in odontocetes in this survey. The presence of *C. grimaldii* in mysticetes, though less frequent [96], was also recorded, with this study marking the first documentation of this metacestode in a fin whale in the region.

Additionally, several other parasites were identified, including the lower-occurring acanthocephalans—eight percent in this survey—such *Bolbosoma vasculosum* and *Bolbosoma capitatum*. *Bolbosoma capitatum* has been documented in large odontocete species, being well known in *G. macrorhynchus* [97,98]. In addition, its presence in *P. electra* is considered in a new report within the Canary Islands, as previously described [98]. In contrast, *Bolbosoma vasculosum* has been reported in a wider range of odontocete species [97,98,99,100,101], reported in this survey in *D. delphis* and *S. coeruleoalba*. Additionally, accidental infections in humans involving parasites of this genus, particularly *Bolbosoma capitatum*, have been documented. The zoonotic potential of these acanthocephalans, similar to that of anisakiasis, is worth highlighting, especially in regions where raw fish consumption is common [37,102,103].

Concerning cyamids, only one specimen could be identified as *Cyamus* sp. Given the limited identification of these parasites, the use of other diagnostic techniques may be useful to confirm the species involved. Furthermore, this study presents the first recorded identification of *Conchoderna auritum* in a *Z. cavirostris* from the Canary Islands, previously reported only in Greek waters [104]. Similarly, *Xenobalanus globicipitis*, a barnacle frequently found in various odontocete species worldwide [97,105,106,107,108,109,110], including those in the Canary Islands, was also observed. These findings broaden the understanding of host–parasite interactions in the region and contribute to the growing body of knowledge on marine mammal parasitology.

Finally, protist cysts of *Sarcocystis* sp. were identified in the muscles of fifteen animals. This prevalence aligns with previous observations, and no inflammatory processes were associated with the presence of these cysts, suggesting that *Sarcocystis* sp. may not have significant pathological relevance [111]. The presence of *Toxoplasma gondii* was confirmed in two *S. coeruleoalba* through molecular and histopathological analyses, making the first reported case of *T. gondii* in *S. coeruleoalba* in the Canary Islands. Prior reports have identified *T. gondii* in *S. frontalis* within the region [5,6,26], where it has been associated with systemic infections and lesions similar to those observed in the current study. *T. gondii* is a notable pathogen and zoonotic protozoan that can cause severe lesions, significantly impacting animal health and potentially leading to stranding and death [26,59,112,113,114,115,116].

The parasitic fauna of cetaceans provides critical insights into host ecology and biology, serving as a valuable tool for understanding the impacts of parasites on cetacean health, which can influence stranding and mortality rates. Accurate parasite identification and health assessment are essential for elucidating imbalances in parasite–host interactions and serve as important biological and environmental indicators. This study has contributed to the knowledge of parasitic species affecting cetaceans stranded in the Canary Islands, including new reports of parasitic species and novel host records, thereby advancing our understanding of the parasitic fauna in this region.

## 5. Conclusions

The present study represents an update of the parasitic species and genera present in cetaceans, both odontocetes and mysticetes, stranded in the Canary Islands over the period between 2018 and 2022. At least eighteen parasitic species and nine genera were identified morphologically in cetaceans during this time.

Significant differences in parasitic prevalence were observed among the analyzed animals, with adult and juvenile/sub-adult animals being the most commonly parasitized, and the genus *Crassicauda*, the most frequently detected (43%).

These findings provide an overview of the current parasitism in cetaceans of the Canary Islands. Collection information on parasite diversity in cetaceans provides an important basis for understanding the impacts on the host and the ecology of the ecosystem. In addition, it allows us to carry out the monitoring of environmental changes over time, offering insights into potential shifts in the marine waters surrounding the Canary Islands, as well as the monitoring of the individual and population health of cetacean populations and their possible impact on human health.

## Figures and Tables

**Figure 1 animals-14-03377-f001:**
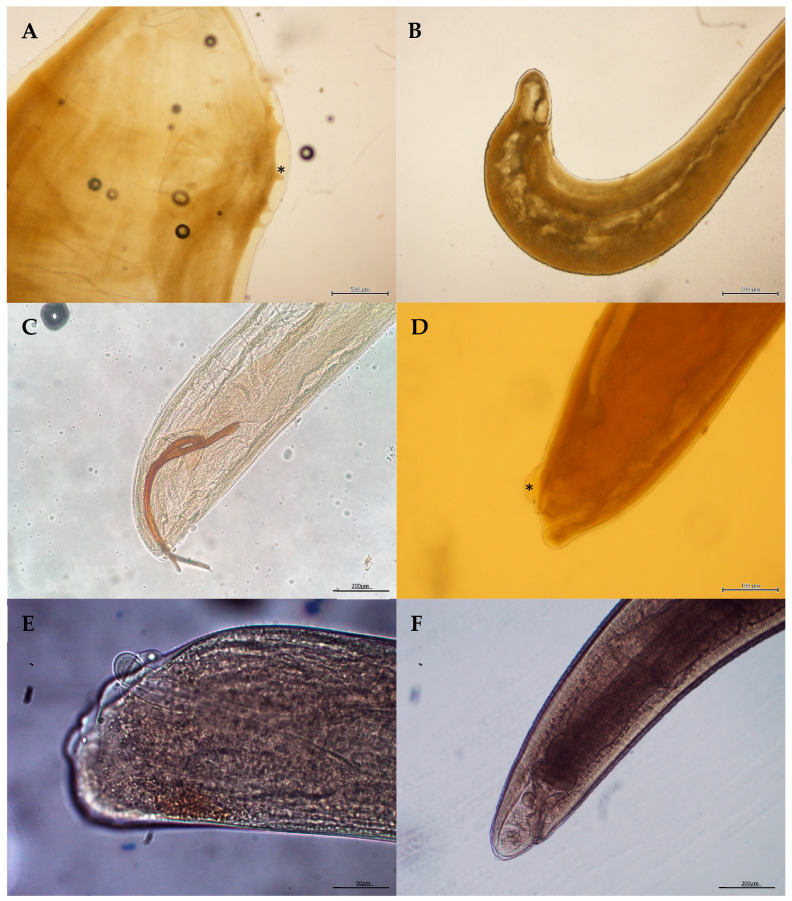
Microscopic views of the nematodes (**A**) Posterior end of a male (papillae in asterisk) of *Crassicauda anthonyi*. (**B**) Posterior end of a male of *Crassicauda grampicola* in the pterygoid sacs of *Grampus griseus*. (**C**) Posterior end of a male of *Halocercus delphini*. (**D**) Posterior end of a female of *Stenurus ovatus* (inflammation of the cuticle in asterisk). (**E**) Posterior end of a female *Stenurus globicephalae*. (**F**) Posterior end of a female of *Anisakis simplex*.

**Figure 2 animals-14-03377-f002:**
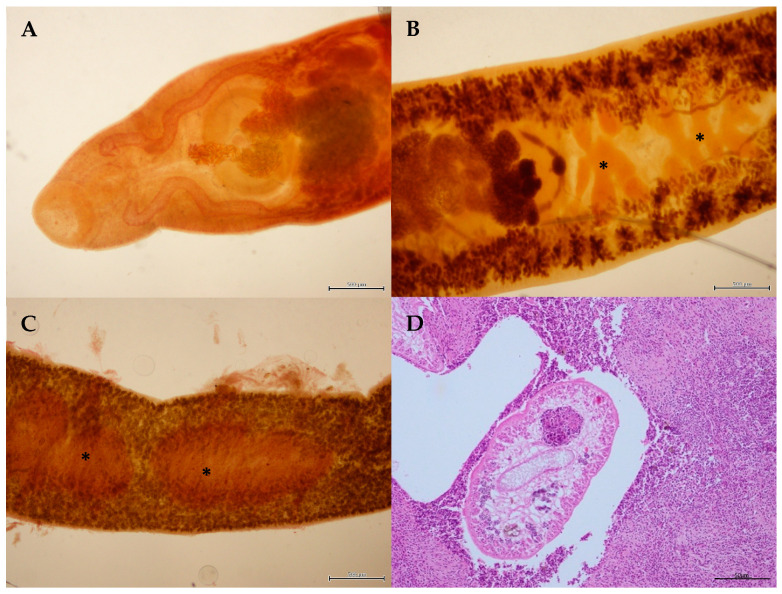
(**A**) Anterior end of *Nasitrema delphini*. (**B**) Detail of the posterior third quarter of the body of *Brachycladium atlanticum* showing the genitalia (testes in asterisks). (**C**) Detail of the middle third quarter of the body of *Oschmarinella rochebruni* showing the testes (asterisks). (**D**) Histological view of *Pholeter gastrophilus* in pyloric stomach leading to granulomatous gastritis. H-E stain.

**Figure 3 animals-14-03377-f003:**
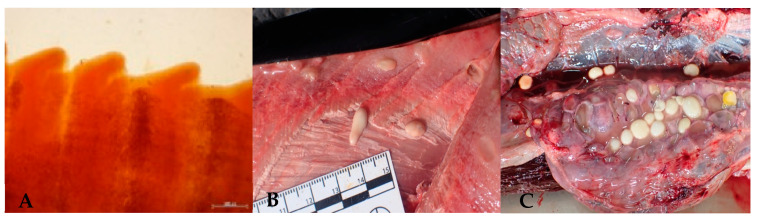
(**A**) Gravid proglottid of *Diphyllobothrium* sp. (**B**) *Clistobothrium delphini* cyst at subcutaneous tissue. (**C**) *Clistobothrium grimaldii* cyst in the abdominal serosa.

**Figure 4 animals-14-03377-f004:**
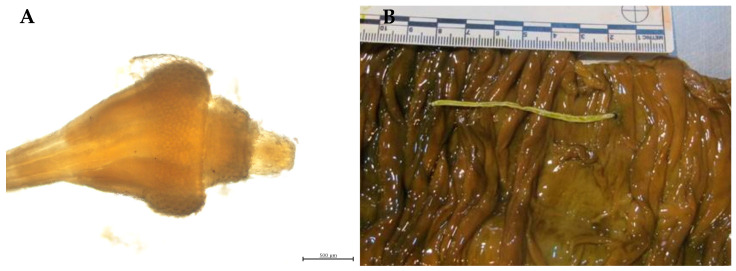
(**A**) Anterior end of *Bolbosoma vasculosum*. (**B**) *Bolbosoma capitatum* embedded in the intestinal mucosa of one short-finned pilot whale during necropsy.

**Figure 5 animals-14-03377-f005:**
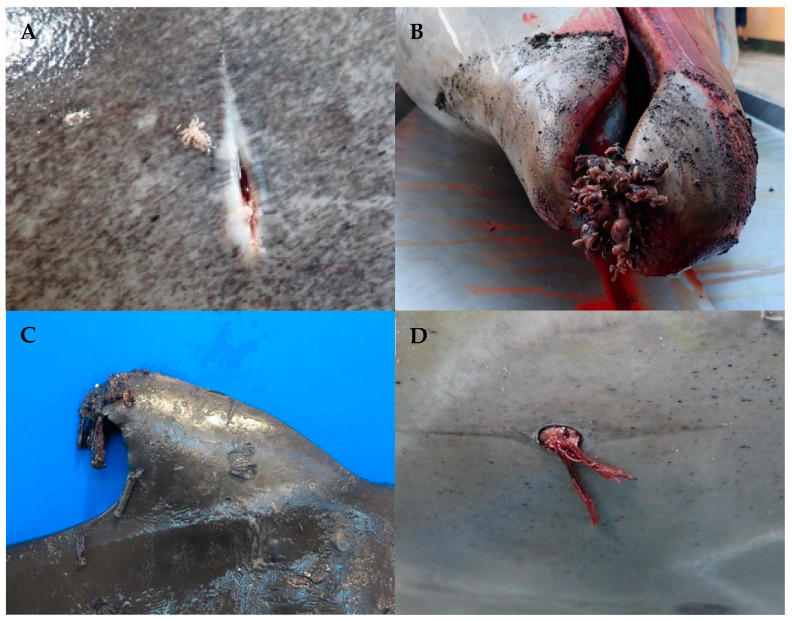
(**A**) *Cyamus* sp. associated with a linear injury in one *Ziphius cavirostris* during necropsy. (**B**) *Conchoderma auritum* attached to the rostral border of the mandible of a *Ziphius cavirostris*. (**C**) *Xenobalanus globicipitis* attached to the dorsal tail in a *Delphinus delphis*. (**D**) *Pennella balaenoptera* associated with ulceration in a *Ziphius cavirostris*.

**Figure 6 animals-14-03377-f006:**
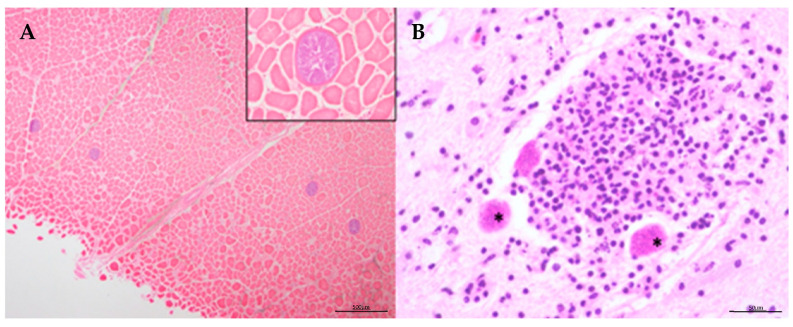
(**A**) Histological views of cysts of *Sarcocystis* sp. in the skeletal muscle. H-E stain. Inset: details of the cyst. x40 H-E. (**B**) Lymphohistiocytic to granulomatous inflammation with some associated *T. gondii* protist cysts (asterisk) of the CNS. H-E stain.

**Table 1 animals-14-03377-t001:** Prevalence of different species according to sex, age, body condition and total prevalence of infected animals stranded in Canary Islands between 2018 and 2022. A: adult; C: calf; F: female; G: good; J-Sad: juvenile–subadult; M: male; Md: moderate; N: neonate; ND: not determined; P: poor; VP: very poor.

	Parasitized Animals (%)	Sex (%)			Age (%)					Body Condition (%)				
		F	M	ND	N	C	J-Sad	A	ND	VP	P	Md	G	ND
*Stenella coeruleoalba* (n = 53/61)	87	40	58	2		11	29	58	2	6	30	47	8	9
*Stenella frontalis* (n = 44/51)	86	52	48		5	29	18	48		2	18	50	14	16
*Delphinus delphis* (n = 25/29)	86	52	48			16	16	68		8	56	16	4	16
*Tursiops truncatus* (n = 15/17)	88	47	53				33	60	7	13	33	27	13	14
*Globicephala macrorhynchus* (n = 13/15)	87	54	46			23	31	46			31	23	23	23
*Grampus griseus* (n = 7/11)	64	57	43			57	29		14	43	14	14		29
*Kogia breviceps* (n = 9/11)	82	33	67			11	33	56						
*Physeter macrocephalus* (n = 4/8)	50	75	25				75	25					50	50
*Ziphius cavirostris* (n = 8/8)	100	50	50				37	50	13	13	13	25	12	37
*Steno bredanensis* (n = 5/6)	83	60	40		20		20	60		20	40	20		20
*Balaenoptera physalus* (n = 1/3)	33		100			100					100			
*Mesoplodon densirostris* (n = 2/3)	67	50	50					100				100		
*Mesoplodon europaeus* (n = 1/3)	33		100					100				100		
*Balaenoptera acutorostrata* (n = 1/2)	50	100						100			100			
*Lagenodelphis hosei* (n = 1/2)	50		100			100						100		
*Balaenoptera edeni* (n = 1/1)	100		100				100						100	
*Kogia sima* (n = 1/1)	100	100						100						100
*Peponocephala electra* (n = 1/1)	100		100				100			100				

**Table 2 animals-14-03377-t002:** Numbers of host individuals parasitized over the number of animals analyzed according to each stranded cetacean species in the Canary Islands. B.a: *Balaenoptera acutorostrata*; B.e: *Balaenoptera edeni*; B.p: *Balaenoptera physalus*; D.d: *Delphinus delphis*; G.m: *Globicephala macrorhynchus*; G.g: *Grampus griseus*; K.b: *Kogia breviceps*; K.s: *Kogia sima*; L.h: *Lagenodelphis hosei*; M.d: *Mesoplodon densirostris*; M.e: *Mesoplodon europaeus*; P.e: *Peponocephala electra*; P.m: *Physeter macrocephalus*; S.c: *Stenella coeruleoalba*; S.f: *Stenella frontalis*; S.b: *Steno bredanensis*; T.t: *Tursiops truncatus*; Z.c: *Ziphius cavirostris*.

Parasites	S.c	S.f	D.d	T.t	G.m	G.g	K.b	P.m	Z.c	S.b	B.p	M.d	M.e	B.a	L.h	B.e	K.s	P.e
Nematodes																		
*Crassicauda* sp.	15/61	31/51	7/29	9/17	4/15	3/11	3/11	-	8/8	-	-	-	-	-	-	1/1	-	-
*Crassicauda anthonyi*	-	-	-	-	-	-	-	-	8/8	-	-	-	-	-	-	-	-	-
*Crassicauda grampicola*	-	2/51	-	1/17	-	4/11	-	-	-	-	-	-	-	-	-	-	-	-
*Halocercus* sp.	-	1/51	-	-	-	-	-	-	-	-	-	-	-	-	-	-	-	-
*Halocercus delphini*	2/61		-	-	-	-	-	-	-		-	-	-	-	-	-	-	-
*Stenurus* sp.	1/61	-	-	-	-	-	-	-	-	-	-	-	-	-	-	-	-	-
*Stenurus ovatus*	-	-	-	4/17	-	-	-	-	-	-	-	-	-	-	-	-	-	-
*Stenurus globicephalae*	-	-	-	-	3/15	1/11	-	-	-	-	-	-	-	-	-	-	-	-
*Anisakis* sp.	20/61	9/51	8/29	4/17	6/15	1/11	8/11	3/8	3/8	2/6	-		1/3	-	1/2	-	1/1	1/1
*Anisakis simplex*	-	-	1/29	-	-	-	-	-	-	-	-	-	-	-	-	-	-	1/1
Trematodes																		
*Nasitrema* sp.	7/61	8/51	9/29	12/17	-	-	1/11	-	-	3/6	-	-	-	-	-	-	-	-
*Nasitrema delphini*	2/61	-	1/29	3/17	-	-	-	-	-	-	-	-	-	-	-	-	-	-
*Brachycladium atlanticum*	3/61	-	1/29	-	-	-	-	-	-	-	-	-	-	-	-	-	-	-
*Oschmarinella* sp.	-	-	-	-	-	-	-	-	2/8	-	-	-	-	-	-	-	-	-
*Oschmarinella rochebruni*	2/61	2/51	1/29	1/17	-	-	-	-	-	1/6	-	-	-	-	-	-	-	-
*Pholeter gastrophilus*	36/61	11/51	11/29	10/17	7/15	1/11	-	-	-	3/6	-	-	1/3	-	-	-	1/1	-
Cestodes																		
*Dyphyllobothrium* sp.	7/61	-	-	-	-	-	-	-	-	-	-	-	-	-	-	-	-	-
*Clistobothrium delphini*	36/61	25/51	12/29	7/17	9/15	1/11	5/11	2/8	6/8	3/6	-	1/3	1/3	1/2		-	1/1	
*Clistobothrium grimaldii*	38/61	11/51	12/29	5/17	7/15	1/11	4/11	-	-	2/6	1/3	-	-	-	-	-	-	-
Acanthocephalans																		
*Bolbosoma vasculosum*	2/61	-	1/29	-	-	-	-		-	-		-	-	-	-	-	-	-
*Bolbosoma capitatum*	-	-	-	-	4/15	-	-	-	-	-	-	-	-	-	-	-	-	1/1
Crustaceans																		
*Cyamus* sp.	-	-	-	-	-	-	-	-	2/8	-	-	-	-	-	-	-	-	-
*Pennella balaenoptera*	2/61	-	-	1/17	-	-	1/11	-	1/8	1/6	-	-	-	-	-	1/1	-	1/1
*Conchoderma* sp.		1/51	2/29	1/17	1/15	-	-	-	2/8		-	-	-	-	-	-	-	-
*Conchoderma auritum*	-	-	-	-	-	-	-	-	1/8	-	-	-	-	-	-	-	-	-
*Xenobalanus globicipitis*	7/61	8/51	11/29	2/17	5/15	4/11	-	-	-	1/6	-	-	-	-	-	-	-	1/1
Protists																		
*Sarcocystis* sp.	4/61	4/51	6/29	-	-	-	-	2/8	-	-	-	-	-	-	-	-	-	-
*Toxoplasma gondii*	2/61	-	-	-	-	-	-	-	-	-	-	-	-	-	-	-	-	-

**Table 3 animals-14-03377-t003:** Endoparasites and ectoparasites reported in the parasitized cetaceans during the period 2018–2022.

Parasites	Host	Site
Nematodes		
*Crassicauda* sp.	*B. edeni*, *D. delphis*, *G. macrorhynchus*, *G. griseus*, *K. breviceps*, *S. coeruleoalba*, *S. frontalis*, *T. truncatus*, *Z. cavirostris*	Mammary gland, Kidney, subcutanean tissue, prostate, pterygoid sacs
*C. anthonyi*	*Z. cavirostris*	Kidney
*C. grampicola*	*G. griseus*, *S. frontalis*, *T. truncatus*	Pterygoid sacs
Non-identified lungworms	*D. delphis*, *G. macrorhynchus*, *G. griseus*, *M. europaeus*, *S. coeruleoalba*, *S. frontalis*, *S. bredanensis*, *T. truncatus*	Lung
*H. delphini*	*S. coeruleoalba*	Lung
*S. ovatus*	*T. truncatus*	Lung
*Stenurus globicephalae*	*G. macrorhynchus*, *G. griseus*	Pterygoid sacs
Non-identified pterygoid sacs nematodes	*D. delphis*, *G. macrorhynchus*, *G. griseus*, *S. coeruleoalba*, *S. frontalis*, *S. bredanensis*, *T. truncatus*	Pterygoid sacs
*Anisakis* sp.	*D. delphis*, *G. macrorhynchus*, *G. griseus*, *K. breviceps*, *K. sima*, *L. hosei*, *M. europaeus*, *P. electra*, *P. macrocephalus*, *S. coeruleoalba*, *S. frontalis*, *S. bredanensis*, *T. truncatus*, *Z. cavirostris*	Digestive tract
*A. simplex*	*D. delphis*, *P. electra*	Digestive tract
Trematodes		
*Nasitrema* sp.	*D. delphis*, *G. macrorhynchus*, *K. breviceps*, *S. coeruleoalba*, *S. frontalis*, *S. bredanensis*, *T. truncatus*	Pterygoid sacs, CNS
*N. delphini*	*D. delphis*, *S. coeruleoalba*, *T. truncatus*	Pterygoid sacs
Non-identified hepatic trematodes	*B. edeni*, *D. delphis*, *M. europaeus*, *S. coeruleoalba*, *S. frontalis*, *T. truncatus*, *Z. cavirostris*	Bile ducts
Non-identified pancreatic trematodes	*D. delphis*, *S. coeruleoalba*, *S. frontalis*, *T. truncatus*	Pancreas
*B. atlanticum*	*D. delphis*, *S. coeruleoalba*	Bile ducts, pancreas
*O. rochebruni*	*D. delphis*, *S. coeruleoalba*, *S. bredanensis*, *S. frontalis*, *T. truncatus*	Bile ducts, pancreas
*P. gastrophilus*	*D. delphis*, *G. macrorhynchus*, *G. griseus*, *K. sima*, *M. europaeus*, *S. coeruleoalba*, *S. frontalis*, *Steno bredanensis*, *T. truncatus*	Stomach
Cestodes		
Non-identified digestive cestodes	*D. delphis*, *G. macrorhynchus*, *G. griseus*, *P. electra*, *S. coeruleoalba*, *S. frontalis*, *T. truncatus*	Digestive tract
*Diphyllobothrium* Cobbold, 1858	*S. coeruleoalba*	Digestive tract
*C. delphini*	*B. acutorostrata*, *D. delphis*, *G. macrorhynchus*, *G. griseus*, *K. breviceps*, *K. sima*, *M. densirostris*, *M. europaeus*, *P. macrocephalus*, *S. coeruleoalba*, *S. frontalis*, *S. bredanensis*, *T. truncatus*, *Ziphius cavirostris*	Blubber
*C. grimaldii*	*B. physalus*, *D. delphis*, *G. macrorhynchus*, *G. griseus*, *K. breviceps*, *S. coeruleoalba*, *S. frontalis*, *S. bredanensis*, *T. truncatus*	Abdominal serous
Acanthocephalans		
Non-identified *acanthocephalans*	*D. delphis*, *G. macrorhynchus*, *M. europaeus*, *P. electra*, *S. coeruleoalba*, *S. frontalis*	Digestive tract
*B. vasculosum*	*D.delphis*, *S. coeruleoalba*	Digestive tract
*B. capitatum*	*G. macrorhynchus*, *P. electra*	
Ectoparasites		
Non-identified cyamids	*D. delphis*, *G. macrorhynchus*, *M. europaeus*, *P. macrocephalus*, *S. coeruleoalba*, *T. truncatus*, *Z. cavirostris*	Skin
*Cyamus* sp.	*Z. cavirostris*	Skin
Non-identified copepods	*B.edeni*, *K.breviceps*, *P.electra*, *S. coeruleoalba*, *S. bredanensis*, *T. truncatus*, *Z. cavirostris*	Skin
*Pennella balaenoptera*	*B. edeni*, *K. breviceps*, *P. electra*, *S. coeruleoalba*, *S. bredanensis*, *T.truncatus*, *Z. cavirostris*	Skin
Non-identified cirripedes	*D. delphis*, *G. griseus*, *P. macrocephalus*, *S. coeruleoalba*, *S. frontalis*, *T. truncatus*	Skin
*Conchoderma* sp.	*D. delphis*, *G. macrorhynchus*, *S. frontalis*, *Z. cavirostris*,	Skin
*C. auritum*	*Z. cavirostris*	Buccal commissure
*Xenobalanus globicipitis*	*D. delphis*, *G. macrorhynchus*, *G. griseus*, *P. electra*, *S. coeruleoalba*, *S. frontalis*, *S. bredanensis*, *T. truncatus*	Skin
Protists		
*Sarcocystis* sp.	*D. delphis*, *P. macrocephalus*, *S. coeruleoalba*, *S. frontalis*	Skeletal muscle
*T. gondii*	*S. coeruleoalba*	Systemic infection

## Data Availability

The data presented in this study are available within the article and the Appendix A.

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
