# Peer review of "Parasitic Infections in Stranded Whales and Dolphins in Canary Islands (2018–2022): An Update"

_animals, 2024, doi:10.3390/ani14233377_

Round 1
Reviewer 1 Report
Comments and Suggestions for Authors
The manuscript “Parasitic infections in stranded whales and dolphins in Canary 2 Islands (2018-2022): An Update” presents relevant information that contributes to the knowledge on the health status of cetacean populations in the Canary Islands. I congratulate the authors for the quality of the manuscript, which meets the scope of the journal and is of interest to its readers. Below, I present some suggestions for modifications and some questions so that the information is better presented.
Line 47 – Canary Islands should be replaced because it is already in the title. I suggest replacing it with “conservation”
Line 86 - The authors did not indicate in the methodology that they evaluated histopathological changes associated with the parasites. However, in figure 2d, the association of Pholeter gastrophilus with granulomatous gastritis was indicated. The histological examination was only mentioned for the identification of Sarcocystis sp. and Toxoplasma gondii cysts (lines 111-112). In lines 304-306 the authors indicate that "Relevant inflammatory responses were observed in the kidneys, muscles, prostate, mammary glands and pterygoid sacs of parasitized cetaceans". How was this analysis performed? The methodology of anatomical-histopathological analysis needs to be described.
Line 103 - check the writing of “phtha2late”
Lines 117 and 119 - change “Sierra et al., 2020” to “Sierra et al. (2020)”
Lines 125 to 134 - I suggest that the species be presented in decreasing numerical order, so that the reader can better visualize the number of species analyzed.
Line 181 - review the description of figure 1a because it does not correspond to the figure, including no spicules or asterisks
Lines 244 to 246 - change “CBW” to “Cuvier’s beaked whale”
Lines 296 to 298 - “Parasite prevalence was similar between males and females, with no apparent association observed between sex and parasite prevalence”. The authors cannot make this claim without a basic statistical analysis.
Line 383 - Accidental parasitism by Bolbosoma capitatum in humans has already been recorded in Japan. It is important that this data be presented and minimally discussed in this study, demonstrating that the zoonotic potential of cetacean parasites is not restricted to T. gondii and Anisakis spp. The possibility of this accidental parasitism in humans (which also occurs with Anisakis spp.) should be indicated throughout the text when the authors indicate the zoonotic potential of some species identified in the study.
Line 384 – change “B. capitacum” to “Bolbosoma capitatum”.
Author Response
|
Comments 1: Line 47 – Canary Islands should be replaced because it is already in the title. I suggest replacing it with “conservation” Response 1: Thank you for the suggestion. We have modified it to ‘conservation’. It can be found on page 1, line 47. Comments 2: Line 86 - The authors did not indicate in the methodology that they evaluated histopathological changes associated with the parasites. However, in figure 2d, the association of Pholeter gastrophilus with granulomatous gastritis was indicated. The histological examination was only mentioned for the identification of Sarcocystis sp. and Toxoplasma gondii cysts (lines 111-112). In lines 304-306 the authors indicate that "Relevant inflammatory responses were observed in the kidneys, muscles, prostate, mammary glands and pterygoid sacs of parasitized cetaceans". How was this analysis performed? The methodology of anatomical-histopathological analysis needs to be described. Response 2: Thank you very much, we agree with this comment. We had not properly addressed this section, as we did not clearly explain the histopathological aspect. Therefore, we have revised Section 2.1 of the Materials and Methods, providing a better explanation of the methodology employed. Additionally, we have explained this section in the cover letter. The correction in the manuscript can be found on lines 104-130, page 3. Comments 3: Line 103 - check the writing of “phtha2late” Response 3: Right, this is a mistake in the typing. It is corrected to ‘phthalate’, line 119, page 3 (Section. 2.2, Matherial and Methods) Comments 4: Lines 117 and 119 - change “Sierra et al., 2020” to “Sierra et al. (2020)” Response 4: I agree with the modification. However, following the style of the citation, we believe that ‘Sierra et al. [31]’ would be correct. We have modified it in lines 128 and 129, page 3, by Sierra et al. [26] Comments 5: Lines 125 to 134 - I suggest that the species be presented in decreasing numerical order, so that the reader can better visualize the number of species analyzed. Response 5: Thanks for the suggestion, We think it could be better visualized this way. The order of the species is modified in decreasing order in lines 135-144, as well as in table 1 and 2. Comments 6: Line 181 - review the description of figure 1a because it does not correspond to the figure, including no spicules or asterisks. Response 6: Thank you very much, you are right. In fact, what you can clearly see in the picture are the parasite papillae, not the spicules. We have changed the caption, line 204 to: ‘Posterior end of a male (papillae in asterisk)’. In addition, We have included the asterisk in figure 1a. Comments 7: Lines 244 to 246 - change “CBW” to “Cuvier’s beaked whale” Response 7: Thank you. I have changed the name to the Latin name in all the document, so the name for CBW's would be Ziphius cavirostris. Lines 273, 274, 276 and others. Comments 8: Lines 296 to 298 - “Parasite prevalence was similar between males and females, with no apparent association observed between sex and parasite prevalence”. The authors cannot make this claim without a basic statistical analysis. Response 8: We agree with this comment. We have removed the part about the association between sex and prevalence of parasitism. Now the sentence would simply read: ‘While, parasite prevalence was similar between males and females’. Line 312, 313. Comments 9: Line 383 - Accidental parasitism by Bolbosoma capitatum in humans has already been recorded in Japan. It is important that this data be presented and minimally discussed in this study, demonstrating that the zoonotic potential of cetacean parasites is not restricted to T. gondii and Anisakis spp. The possibility of this accidental parasitism in humans (which also occurs with Anisakis spp.) should be indicated throughout the text when the authors indicate the zoonotic potential of some species identified in the study. Response 9: Thank you very much, it is a very interesting comment. We have included a paragraph and the corresponding references. The paragraph is on lines 404-407, page 14. Comments 10: Line 384 – change “B. capitacum” to “Bolbosoma capitatum”. Response 10: Thank you. We have changed it, line 400, page 14.
|

Reviewer 2 Report
Comments and Suggestions for Authors
The manuscript submitted to the journal is based on long-term research. It is well structured and well written. I have only a few questions and suggestions for improving it.
1) The introduction is very short. I propose to expand it a little by publishing about possible ways of transferring parasites to humans.
2) Make calculations of the intensity and extent of infection.
3) Completely redo the conclusion. Now this conclusion does not correspond to the results obtained. It is also necessary to show the practical significance of the results here.
Author Response
|
Comments 1: The introduction is very short. I propose to expand it a little by publishing about possible ways of transferring parasites to humans. Response 1: Thank you very much for your appreciation. We have modified the introduction, including some important aspects of the role of cetaceans in the marine environment, as well as their importance in zoonotic role. Comments 2: Make calculations of the intensity and extent of infection Response 2: Thank you very much for your comments, we agree that it is important to include the intensity and extent of the infection. However, since a quantitative estimation of the infection could not be made, it is not possible to discuss intensity in this case. Therefore, an attempt was made to give a subjective value to the infection -mild, moderate and severe infection- in a way that tries to reflect the extent of parasite damage. In future studies we would like to carry out a more exhaustive study taking into account the intensity and severity of the lesions. Comments 3: Completely redo the conclusion. Now this conclusion does not correspond to the results obtained. It is also necessary to show the practical significance of the results here. Response 3: Thanks for the suggestion. We have redrafted the conclusions, adjusted them more in accordance with the results obtained.
|

Reviewer 3 Report
Comments and Suggestions for Authors
Whale and dolphin strandings are a global problem. The manuscript under review deals with endo and ectoparasites recorded in stranded cetaceans in Canary Islands in 2018-2022. The parasite fauna of cetaceans can serve as an indicator for understanding host biology and the impact of parasites on the health of whales and dolphins. The authors presented very interesting data, which are well illustrated with photographs of the parasites found. The article presents new data on parasites and their cetacean hosts in the Canary Islands. The manuscript undoubtedly meets purposes and objectives of the Animals and can be published.
But I have some remarks about this manuscript:
Lines 34,35 - So who had the highest infection rate - young or adult animals? The sentence needs to be rephrased.
Regarding Table 1. It is more suitable for the Material and Methods section, since it describes the research material more.
Section 3.2.1. – Tables 2 and 3 should be placed immediately after their first mention in the text.
In table 3, I would suggest that the authors change the names of columns 2 and 3, simply: Host and Site. And the table name could be simpler: “Endoparasites and ectoparasites reported in the parasitized cetaceans during the period 2018-2022”.
Lines 102, 105 - It seems to me that the methods are not described by these authors. Is the link wrong? It is necessary to check the correctness of all references.
Line 135 - the link is incorrect - there are no “respective epidemiological data” in S1 Table. In general, what do the authors mean by the respective epidemiological data? Apparently, the name of the table needs to be changed.
Please note and correct throughout the text - the correct name of cestode genus is Diphyllobothrium Cobbold, 1858
Lines 147, 148 and further – In scientific articles it is better to use Latin names of animals. Try to avoid common names.
According International Code of Zoological Nomenclature (ICZN) at the first mention of genera or species in the article text its full Latin name with the author and year of description should be given; in relation all species of living organisms – parasites and their hosts (lines 72,73: Anisakis Dujardin, 1845; Toxoplasma gondii (Nicolle & Manceaux, 1908), section 3.1., etc.). This should be done for all species and genera in Latin in the article at the first mention.
Authors must submit the manuscript in accordance with the rules for formatting MDPI articles: article title and the names of subsections, all words in which must be capitalized.
Lines 52,53 – This is a pun. … including …. including. Please, rephrase.
Line 80 – better write – “… have been identified” or “known”
Lines 117, 119- correct reference - Sierra et al. [31].
Line 375 – must be rephrase like this: “Diphyllobothrium spp. is the only zoonotic cestodes known …”
Line 421 – It would probably look better like this: “Parasites of nine genera …”
The manuscript can be published, but some corrections are needed.
Author Response
|
Comments 1: Lines 34,35 - So who had the highest infection rate - young or adult animals? The sentence needs to be rephrased. Response 1: Thank you for your comment. Adults and sub-adults/juvenile animals were the groups with the highest prevalences, compared to calves and neonates. We have rephrased the sentence in line 34-35, page 1. Comments 2: Regarding Table 1. It is more suitable for the Material and Methods section, since it describes the research material more. Response 2: Thank you very much, we have considered this fact, and we agree. However, the data presented show also prevalence values, within the parasitized animals, so we believe that they would enter as results as well. However, we have also referenced table 1 in the material and methods section. Line 97, page 3. Comments 3: Section 3.2.1. – Tables 2 and 3 should be placed immediately after their first mention in the text. Response 3: Thank you very much. We have modified the position of the tables on pages 5-7. Comments 4: In table 3, I would suggest that the authors change the names of columns 2 and 3, simply: Host and Site. And the table name could be simpler: “Endoparasites and ectoparasites reported in the parasitized cetaceans during the period 2018-2022”. Response 4: Thank you, we appreciate your suggestions. We have modified the title of the table and included the words host and site in Table 3. Comments 5: Lines 102, 105 - It seems to me that the methods are not described by these authors. Is the link wrong? It is necessary to check the correctness of all references. Response 5: Thank you very much for your comment. Regarding the methodology carried out in this work, we followed the same procedure as Fraija-Fernandez et al., 2014, in their methodology section ‘Morphological analyses’. They use the following: ‘Seventeen specimens were stained with iron acetocarmine and differentiated using HCl in 70% ethanol. Specimens were dehydrated through a graded ethanol series, cleared with dimethyl phthalate and mounted as permanent preparations in Canada balsam’. This is why this paper has been referenced in this article. Comments 6: Line 135 - the link is incorrect - there are no “respective epidemiological data” in S1 Table. In general, what do the authors mean by the respective epidemiological data? Apparently, the name of the table needs to be changed. Response 6: Thank you for your appreciation. The location of the stranding had been entered as epidemiological data. Now, we have modified the title of the table to simplify it to “biological data and stranding location”. In addition, we have modified the title of sections 2.1 and 3.1 to ‘Necropsied animals and biological data’ , lines 93 and 130. Comments 7: Please note and correct throughout the text - the correct name of cestode genus is Diphyllobothrium Cobbold, 1858. Response 7: Thank you very much for your appreciation. The correct name of the genus is already included throughout the document (lines 238 and 390). Comments 8: Lines 147, 148 and further – In scientific articles it is better to use Latin names of animals. Try to avoid common names. Response 8: Thank you for the suggestion. We’ve modified the common names of animals for the Latin names throughout the document. You can find it in lines 159 and further. Comments 9: According International Code of Zoological Nomenclature (ICZN) at the first mention of genera or species in the article text its full Latin name with the author and year of description should be given; in relation all species of living organisms – parasites and their hosts (lines 72,73: Anisakis Dujardin, 1845; Toxoplasma gondii (Nicolle & Manceaux, 1908), section 3.1., etc.). This should be done for all species and genera in Latin in the article at the first mention. Response 9: Thank you for your appreciation. The author's description and the year accompanying the genera and species are included from page 2, line 65 onwards, as they are named for the first time during the document. Comments 10: Authors must submit the manuscript in accordance with the rules for formatting MDPI articles: article title and the names of subsections, all words in which must be capitalized. Response 10: Thank you for your comment. We have checked the instructions for authors and the ‘Animals’ template and it does not indicate that the title has to be capitalized. If you could confirm that the title and subsections will be in capital letters after publication, we will rectify this immediately. Comments 11: Lines 52,53 – This is a pun. … including …. including. Please, rephrase. Response 11: Thank you for your appreciation. We have changed the introduction and the sentence, Lines 52-54, page 2. Comments 12: Line 80 – better write – “… have been identified” or “known” Response 12: Thank you for your appreciation. The phrase is now written as: “have been identified”. Line 86, page 2. Comments 13: Lines 117, 119- correct reference - Sierra et al. [31]. Response 13: Thank you very much, they are now adjusted to the format of the citation on lines 128-129. Comments 14: Line 375 – must be rephrase like this: “Diphyllobothrium spp. is the only zoonotic cestodes known …” Response 14:Thank you for your correction. We have rephrased the sentence in line 390. Comments 15: Line 421 – It would probably look better like this: “Parasites of nine genera …” Response 15: Thank you for your comments. We have modified the conclusions of the article, re-drafted them in a way that better reflects the results. |
